# Life cycle assessment needs predictive spatial modelling for biodiversity and ecosystem services

Rebecca Chaplin-Kramer[1], Sarah Sim[2], Perrine Hamel[1], Benjamin Bryant[1], Ryan Noe[3], Carina Mueller[2], Giles Rigarlsford[2], Michal Kulak[2], Virginia Kowal[1], Richard Sharp[1], Julie Clavreul[2], Edward Price[2], Stephen Polasky[3,4], Mary Ruckelshaus[1] & Gretchen Daily[1,5]

International corporations in an increasingly globalized economy exert a major influence on the planet's land use and resources through their product design and material sourcing decisions. Many companies use life cycle assessment (LCA) to evaluate their sustainability, yet commonly-used LCA methodologies lack the spatial resolution and predictive ecological information to reveal key impacts on climate, water and biodiversity. We present advances for LCA that integrate spatially explicit modelling of land change and ecosystem services in a Land-Use Change Improved (LUCI)-LCA. Comparing increased demand for bioplastics derived from two alternative feedstock-location scenarios for maize and sugarcane, we find that the LUCI-LCA approach yields results opposite to those of standard LCA for greenhouse gas emissions and water consumption, and of different magnitudes for soil erosion and biodiversity. This approach highlights the importance of including information about where and how land-use change and related impacts will occur in supply chain and innovation decisions.

[1] Natural Capital Project, Woods Institute for the Environment, Stanford University, Stanford, California 94305, USA. [2] Unilever Safety and Environmental Assurance Centre, Unilever R&D, Colworth Science Park, Sharnbrook, Bedfordshire MK44 1LQ, UK. [3] Natural Capital Project, Institute on the Environment, University of Minnesota, St Paul, Minnesota 55108, USA. [4] Department of Applied Economics, University of Minnesota, St Paul, Minnesota 55108, USA. [5] Department of Biology, Center for Conservation Biology, Stanford University, Stanford, California 94305, USA. Correspondence and requests for materials should be addressed to R.C.-K. (email: bchaplin@stanford.edu).

The size and reach of multinational companies are on par with that of many nations, making private sector sustainability commitments crucial to decoupling environmental impact from economic growth[1]. Meaningful commitments involve comprehensive shifts in product design, resource consumption and sourcing strategies[2]. The need to assess the environmental impacts of such shifts—impacts on land use, biodiversity, climate, water and key pollutants[3]—is driving rapid innovation in science and tools for transforming decisions in private and public sectors[4].

Life cycle assessment (LCA) provides a framework for companies to evaluate the impacts of product and supply chain decisions. Recent methodological advances extend the range of environmental impacts considered to include biodiversity and ecosystem services[5–7]. The application of LCA remains limited in key ways, however, when considering land-use change. First, the currently applied methods for estimating land-use change are based on linear extrapolation of past changes at the country level[8]. Second, land within nations or ecoregions is homogeneous, with spatial differentiation only across such units[9]. Third, landscape configuration (for example, habitat fragmentation) and context (for example, proximity to landscape features such as watercourses) are ignored[10], even though local spatial heterogeneity and landscape configuration are key factors in determining impacts on biodiversity and many ecosystem services[11,12]. Finally, a predictive framing typically only used in consequential (change-based or marginal) LCA would also benefit attributional (accounting' or 'non-marginal) LCA[13] so that total changes in company and sectoral decision-making can be anticipated[14]. (For more information on the distinction between attributional and consequential LCA, see Supplementary Note 1; also ref. 15.)

Three key steps are needed to connect supply chain decisions to on-the-ground environmental impacts in a way that can promote more sustainable raw material production. First, companies must understand where in the world their additional demand for a product will result in increased production. Second, increased production must be translated into a change in land use (either intensification or expansion). Finally, the impacts resulting from that change in land use must be meaningfully assessed. Tremendous progress has been made and much effort is still underway, both to spatialize current sourcing locations[16–18] and to predict future production patterns[19]. We recognize this first step as critical to full application in decision-making. However, the focus of this paper is to link steps two and three into existing LCA techniques and reveal the importance of a spatially explicit approach in understanding the full impacts of production in a given sourcing region.

We demonstrate here how globally available, spatial data and newly accessible tools for ecosystem services can be applied to predictive modelling of large-scale changes in agricultural systems through LCA. We substitute key elements of life cycle inventory in the agricultural stage of an attributional LCA with outputs from predictive land-change modelling (LCM) and spatially explicit ecosystem services modelling using the InVEST software suite[20]. We call our new approach Land-use Change Improved (LUCI)-LCA (Fig. 1 and Supplementary Methods). We apply this approach to explore the implications of large-scale growth in demand for bio-based, high-density polyethylene (HDPE). We analyse three production volume scenarios (23,000, 86,000 or 321,000 tonnes), which represent the types of changes that could arise from a large company or sector-level shift in demand. Further, we compare two bio-feedstocks of bio-based HDPE (maize grown in Iowa, USA and sugarcane grown in Mato Grosso, Brazil) to understand how environmental impacts differ based on feedstock and location. We compare results using the new LUCI-LCA approach to the standard attributional, ISO4040-compliant LCA (hereafter 'standard LCA') for five LCA impact categories relevant for agricultural products (Fig. 1), and find impacts of different order and magnitude between the two methods.

## Results

**Rank-order and relative differences between commodities**. For global warming potential and water consumption, LUCI-LCA reveals strikingly different results to those from standard LCA, resulting in a different bio-feedstock preference for decisions seeking to minimize these impacts related to bio-plastic production (Fig. 2). For the other impact categories, the two methods agree in the direction of the differences between feedstocks, but the magnitude of these differences varies substantially (the difference in impacts predicted by LUCI-LCA is less than half that of standard LCA for biodiversity damage potential and more than three times that for erosion potential). We focus on these relative differences rather than comparing absolute values between the two methods because LCAs are most often used to compare the environmental impacts of alternative options, rather than for mitigating or valuing impacts. Furthermore, the major methodological differences introduced in taking a predictive, spatially explicit approach mean that the more meaningful interpretation is not whether the methods produce different results (they do because they are measuring different things), but whether they might induce different choices in the decision context.

Indeed, the methodological differences between standard LCA and LUCI-LCA underlie the differences in rank order and magnitude of impacts between the two feedstocks and regions. For global warming potential, standard LCA assigns carbon losses in proportion to the current composition of land cover; the higher carbon loss in Brazil reflect the much higher proportion of forest in Mato Grosso compared to Iowa. In contrast, the LCM in LUCI-LCA suggests that most agricultural expansion would occur on savanna in Mato Grosso, and a larger proportion on forest in Iowa, thus predicting greater carbon losses in Iowa (Supplementary Fig. 4). While the InVEST model accounts for spatial dynamics in carbon storage in tropical forests[21], these have minimal impact on the results since only a small amount of forest is indicated for conversion in Mato Grosso. For water consumption, methods are consistent between LUCI-LCA and standard LCA, but different data sources and processing (for example, for climate, land use and yields) result in markedly higher impacts for sugarcane relative to maize for LUCI-LCA.

For the other impacts, it is the ecosystem service modelling and LCM together that produce the different results for LUCI-LCA. Biodiversity damage potential in standard LCA accounts for the difference between the potential natural vegetation in a region and its current occupied state[7], while in LUCI-LCA the difference between the current (2007) and modelled future landscape is considered. In addition, LUCI-LCA accounts not only for biodiversity changes on land that is converted but also for changes on adjacent land (due to fragmentation), which yields relatively higher biodiversity impacts for Iowa because of the limited natural land remaining there (resulting in less of a difference between sugarcane and maize than when using standard LCA approaches). For erosion potential, standard LCA estimates the total soil erosion on land[6], while LUCI-LCA estimates sediment exported to water, accounting for the buffering role of vegetation that retains sediment before it reaches watercourses. This explains why the conversion of last remaining riparian habitat in Iowa creates such high impacts

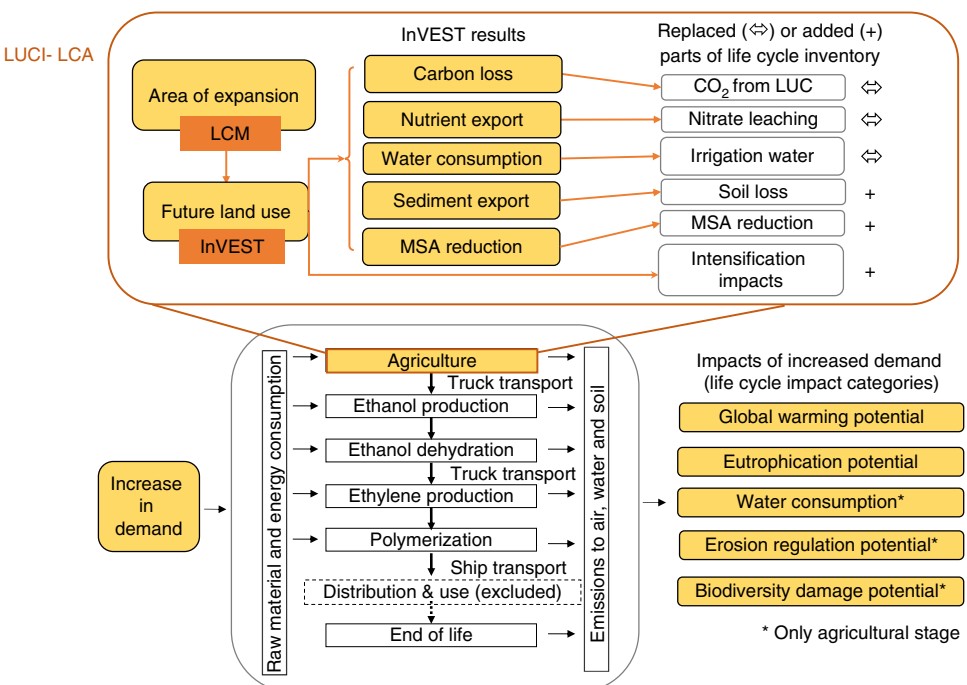

**Figure 1 | Conceptual framework for LUCI-LCA.** Two key innovations to standard LCA are shown in orange; the elements of the life cycle that are modified by these innovations are highlighted in yellow. First, land-change modelling (LCM) based on logistic regression with climatic and soil suitability is used to forecast plausible future agricultural expansion (including intensification), rather than attributing future footprint based on current status as done in standard attributional LCA. Second, land-use change (LUC) is translated to impacts using spatially explicit models for biodiversity (MSA) and ecosystem services (InVEST), rather than assuming linear relationships between impacts and crop production as in standard LCA. All changes to inventories and impacts occur within the agriculture stage of the life cycle, shown in expanded form (in the orange box) above the full standard LCA schematic in grey.

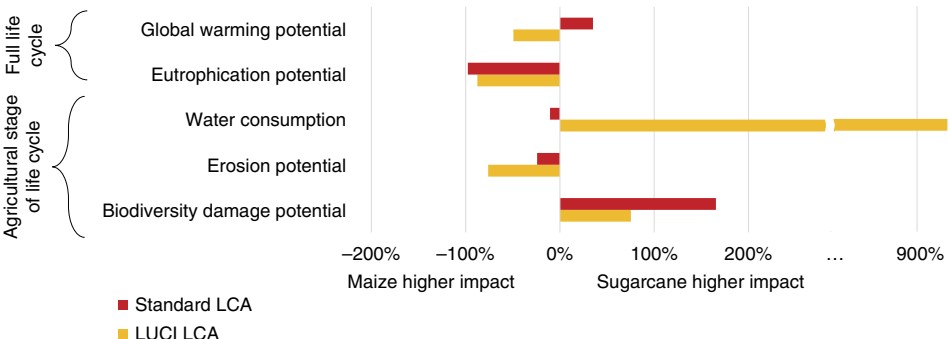

**Figure 2 | Relative differences in impacts of HDPE produced from sugarcane versus maize using LUCI-LCA and standard LCA.** Relative differences are displayed as the percent by which sugarcane-based HDPE has higher (positive values) or lower (negative values) impact than maize-based HDPE for the same production level. Red bars are calculated using standard LCA methodology (constant for all production scenarios); yellow bars using LUCI-LCA (for scenario 3 = 321,000 tonnes HDPE).

relative to those in Mato Grosso. Likewise, the LUCI-LCA methods for eutrophication potential include buffering effects of vegetation in the landscape, although in this case, standard LCA assumptions about tile drainage (in Iowa and not Mato Grosso) mimicked this effect.

**Nonlinear effects of increasing scale of production.** The rank order of impacts from the two bioplastics is retained for all three production volume scenarios, despite the fact that impacts in LUCI-LCA can change with the scale of production (Fig. 3). The differences in impacts between these two feedstocks tend to increase with higher volumes of plastic, especially for global warming potential (ranging from a 14% difference between the two feedstocks for 23,000 tonnes of HDPE to 40% for 321,000 tonnes) and biodiversity damage potential (from 20 to 75%). However, it is worth noting that this may not always be the case; the nonlinearity of impacts means that one feedstock could have lower impacts than the other at low production scenarios, but higher impacts than the other under higher production scenarios. Such nonlinearities are highly relevant for comparative decision contexts where new technologies or materials like bio-based plastics are rolled-out in a stepwise manner, with total volumes anticipated to be much larger than initial market penetration.

**Estimated impacts from LUCI-LCA are robust to uncertainty.** The rank order of impacts from the two bioplastics is maintained even with uncertainty resulting from model parameter and LCA yield assumptions (error bars in Fig. 4, based on upper and lower bounds of sensitivity analysis; Supplementary Table 29), for three of the five impact categories considered. This is true for standard LCA as well, meaning that the differences exhibited between the two methods are consistent even when accounting for this

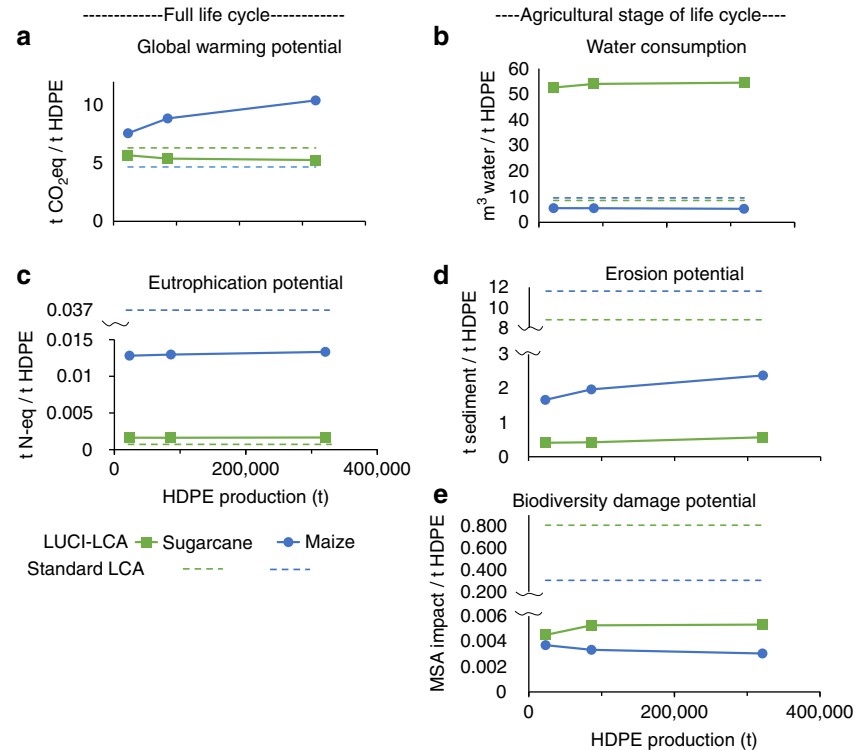

**Figure 3 | Non-linear impacts from increasing scale of production in LUCI-LCA.** Three scenarios of production impacts per tonne of HDPE production, on (**a**) global warming potential, (**b**) water consumption, (**c**) eutrophication potential, (**d**) erosion potential and (**e**) biodiversity damage potential, for sugarcane (in green) and maize (in blue). Solid lines show change in impacts with production volume according to LUCI-LCA. Standard LCA estimates are provided for reference (dotted lines), showing constant per tonne impacts regardless of production amounts.

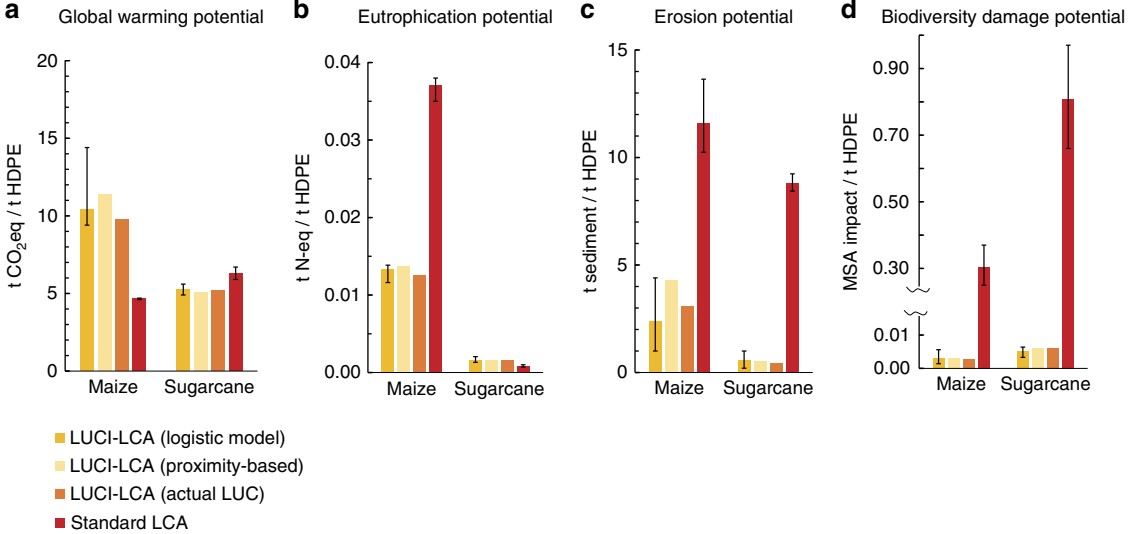

**Figure 4 | Comparison of impacts resulting from different assumptions of land-use change under increased bio-feedstock demand.** LUCI-LCA impacts of HDPE production on (**a**) global warming potential, (**b**) eutrophication potential, (**c**) erosion potential and (**d**) biodiversity damage potential. Impacts modelled using the logistic LCM (in dark yellow) and proximity-based LCM (in light yellow) are presented per tonne of HDPE for production scenario 3 (321,000 tonnes). LUCI-LCA impacts corresponding to actual LUC (in orange) are the impacts modelled in LUCI-LCA for agricultural expansion that occurred between 2007 and 2012 in both regions (Iowa for maize and Mato Grosso for sugarcane), converted to per tonne of HDPE production possible if that amount of additional agricultural acreage were used for HDPE production from the respective bio-feedstocks. Standard LCA results (in red) are shown as the per-tonne impact for every scenario (all identical, as shown in Fig. 2). Error bars show the high and low estimates of sensitivity analysis resulting from uncertainty in InVEST parameter and crop yields (Supplementary Methods and Supplementary Table 29). LUC, land-use change.

uncertainty. However, for biodiversity damage potential the difference in impacts between feedstocks is much smaller for LUCI-LCA than for standard LCA (Fig. 2), such that the lower-bound estimates for sugarcane overlap with the upper-bound estimates for maize (Fig. 4). For water consumption (not shown in Fig. 4; see Supplementary Table 29), we find the reverse, with widely overlapping bounds of impact for the two feedstock crops in standard LCA (1.8–11.5 m$^3$ water per tonne HDPE for sugarcane; 1.6–17.5 for maize) but consistently higher impacts for sugarcane in LUCI-LCA, despite considerable uncertainty (0.9–9.7 m$^3$ water per tonne HDPE for maize; 11.7–73.7 for sugarcane).

The differences in impact between the two feedstocks are also robust to uncertainty in spatial predictions of land-use change, as evidenced by a comparison of LCM impacts to those from agricultural expansion that took place between 2007 and 2012 (Fig. 4). The impacts of this historical land-use change are estimated by applying the same InVEST models and subsequent LUCI-LCA approach. This serves as a form of validation for the effect of modelled land-use change on each impact category (except water consumption; Supplementary Methods). The LUCI-LCA estimate (logistic model; Fig. 4) comes closer to matching the modelled impacts of actual land-use change than does standard LCA for all impacts. Even a much simpler LCM, applying the same HDPE production scenarios, but assigning agricultural expansion suitability based on proximity to current agricultural land (proximity-based; Fig. 4), performs nearly as well as the logistic LCM in LUCI-LCA.

## Discussion

The strength of LUCI-LCA is in illuminating the importance of location in predicting impacts of future land use, rather than relying on regional averages of current land use. These findings, revealing differences not only in the magnitude of impacts but also in the ranking of feedstock options, are important for multinational companies and other global actors driving shifts in agricultural product supply chains. Increased demand for a given crop from a certain location will result in changes to production, in terms of both intensification and expansion, which may ultimately (even if indirectly, through displacement of other crops) spill over onto natural areas[22]. Many ecosystem processes depend on landscape configuration in the context of topography, climate, soils and so on[23], such that disregarding these spatial elements can lead to erroneous conclusions about the impacts of a particular decision.

The LUCI-LCA approach improves the spatial resolution and ecological processes represented in LCA, which can feed into or further enhance the continuing evolution of LCA science. Recent advances in life cycle impact assessment methods provide regional characterization factors for biodiversity using a 'country-side species–area relationship'[9]. However, life cycle inventories of agricultural raw materials that provide building blocks for LCA studies are still generated without consideration of local landscape configuration. The spatially explicit MSA measure derived in LUCI-LCA addresses this major modelling gap. In addition, the species–area relationship method[9] considers only the number of species being lost, while MSA includes species abundance, an improvement suggested by recent consensus[24]. Meanwhile, advances in water footprinting methodology, such as AWARE[25], allow for consideration of water availability to contextualize the impact of increases in water consumption. The AWARE characterization factors for the two case study regions considered suggest a slightly greater proportion of available water remaining in Mato Grosso compared to Iowa. However, when weighting our water consumption results by these characterization factors, our conclusions do not change:

sugarcane from Mato Grosso still has a much higher impact on water than maize from Iowa (Supplementary Note 6).

Importantly, the advances presented here can be incorporated into mainstream LCA or environmental impact assessment more generally. The LUCI-LCA method can be applied with globally available data and open-source, free tools. This is critical for broad uptake, since supply chains often span many nations and whole sectors of production, and thus most product design or procurement decisions have global impact[26]. At a minimum, even a simple LCM based on proximity to current agriculture marks a major step forward in predicting impacts. The integration of spatially explicit ecosystem service models such as InVEST can reveal unexpected potential consequences resulting from the loss of habitat that plays a buffering or connective role in mediating impacts. Weighting impacts by their relative importance in regions of interest (based on beneficiaries of the services, or other factors) would further enhance this approach, helping decision-makers to evaluate trade-offs between different environmental impact categories. For water consumption, such methods are already advancing through the development of water scarcity metrics for LCA[27].

Our approach illustrates the different estimates of impacts that spatially explicit, predictive models can yield for particular sourcing locations, but this points to a major research and information gap concerning supply chain transparency. While we chose two possible sourcing regions for the two feedstocks in this example (Iowa and Mato Grosso), supply chains are typically complex, and companies are often unlikely to know exactly where new commodity demand translates to increased production on the ground. Recent efforts to downscale country-to-country trade analyses can provide subnational commodity origin probabilities for different countries[16,18], and such spatially explicit trade mapping has already been shown to result in differences of more than 20% when accounting for virtual water use[17]. These advances could help to define probabilistic production scenarios for a LUCI-LCA analysis, and enable more realistic projections of where and how corporate supply chain decisions would impact biodiversity and ecosystem services.

While spatial understanding of trade flows would be a critical first step towards linking the fine-scale spatial LCM to the supply response of an increase in demand for agricultural feedstocks, the ability to anticipate future changes in production patterns hinges on economics-based approaches such as equilibrium modelling[28–30] or supply cost modelling[31,32]. For such models to be practical, however, they need to be integrated in ways that promote transparent, global assessment with the ease of use that the LUCI-LCA approach offers. Mainstream equilibrium models (for example, GTAP[28] and GLOBIOM[30]) require expertise and licensing fees that deter use by many LCA practitioners; they may also lack granularity in the way they represent the agricultural sector.

Companies need reliable approaches that are fit for purpose to inform strategic innovation and sourcing decisions. Translating state-of-the-art understanding of ecosystem impacts into decision-ready information requires predictive, system-scale, robust modelling that allows rapid assessment with accessible tools. LCA already provides a means towards meeting those needs, reflected in its increasingly broad uptake. As currently applied, however, LCA may misrepresent the impacts of the decisions it is intended to inform, if it ignores important ecosystem processes and spatial heterogeneity. Such spatially explicit, predictive information about the consequences of decisions for biodiversity and ecosystem services has been applied in agricultural policy contexts, such as the design of incentives programmes for best-management practices or prioritization for conservation activities in agricultural landscapes[33]. Scientists and practitioners can build on the momentum generated by these and

other examples, and the promise held by many emerging corporate sustainability commitments, by equipping companies with the information and tools they need to realize the full potential of those commitments.

## Methods

**Overview.** To estimate the environmental impacts of different feedstocks and locations, and to ascertain the degree to which spatially explicit data and methods change the results, we conduct an LCA for HDPE bioplastic using two alternative approaches (Supplementary Methods). After defining the demand and therefore feedstock production scenarios (Supplementary Table 1), we first conduct a standard attributional, ISO4040 compliant LCA (hereafter 'standard LCA'; Supplementary Figs 1 and 2), calculating land-use change (transformation) impacts with the direct land-use change assessment tool developed by Blonk Consultants. This represents historical changes in the land cover for the given crops (sugarcane and maize) and country of production (Brazil and the USA) over the last 20 years. We also apply our newly developed LUCI approach to LCA (Fig. 1), in which we use the results of spatially explicit modelling of land change and its impacts on biodiversity and ecosystem services to inform the agricultural stage of the standard LCA. To illustrate the impact of scale on sourcing decisions, we apply each approach to a range of demand scenarios for bioplastic crop production and the land-use changes required to meet them.

The LUCI-LCA is based on attributional inventories but considers forward-looking expansion and intensification on new land, based on predictive spatial modelling to meet demand for the new material. We assume the new demand is additional to maintaining existing agricultural production levels for current uses. The modelling takes into account historical trends for both intensification of production on existing land and suitability of land for expansion in the area. Below, and in more detail in the Supplementary Methods, we discuss the various elements of our approach: (1) definition of demand scenarios; (2) elements of the standard LCA; (3) elements of the LUCI-LCA approach, including predictive LCM, biodiversity and ecosystem service modelling; and (4) integration of LUCI into LCA, including adaption of existing life cycle inventory used in the standard LCA.

**Demand scenarios.** Recognizing the importance of geographical influences on the results of such assessments, two different feedstocks in two locations are considered to demonstrate the new LUCI-LCA approach. Three different volumes of feedstock for differing bio-HDPE demand scenarios are explored: 23,000; 86,000; and 321,000 tonnes. The first two volume scenarios are set at scales that could be induced directly by Unilever, with a subsequent scenario set to represent broader sectorial uptake of the bio-HDPE. Scenario 1 (23,000 tonnes) represents Unilever's approximate HDPE packaging volume used in North America in 2012 and assumed to be met either from maize grown in the USA or sugarcane from Brazil. Scenario 2 (86,000 tonnes) represents Unilever's approximate total plastic packaging volume (all plastics, that is, PP + PET + HDPE) used in North America in 2012. This total volume was considered as HDPE in the case study to give a sense of the impacts that would emerge if they were sourced from bio-feedstocks. Scenario 3 (321,000 tonnes) is an extrapolated volume included to scale the demand through the sector (calculated by taking the ratio of Unilever HDPE volume and Unilever plastic volume in North America and multiplying by the total plastic volume in North America), met either from maize grown in the USA or sugarcane from Brazil.

**Standard LCA.** The standard LCA conforms to prevailing methods for the selected impact categories, as follows. Global warming potential uses the IPCC AR5 (ref. 34) method for 100 years excluding biogenic carbon. Eutrophication potential uses the ReCiPe method[35], water consumption inventory data originate from a database from the Water Footprint Network[36] (Supplementary Table 2) and erosion regulation potential is computed according to Saad et al.[6] Biodiversity damage potential follows de Baan et al.[37], using mean species abundance (MSA)[38] to estimate impacts on biodiversity.

The amount of land-use change (transformation) and carbon dioxide emissions resulting from land transformation are estimated using the direct land-use change assessment tool[39], following the PAS2050-1 (ref. 8), Greenhouse Gas Protocol[40] and EnviFood Protocol[41], according to the 'country known, previous land use unknown' situation. The area of land transformation used in the assessment of erosion potential and biodiversity damage potential are given in Supplementary Table 3 and the greenhouse gas emissions from land transformation for sugarcane and maize are 10.81 tonne $CO_2$eq per hectare per year and 0.02 tonne $CO_2$eq per hectare per year, respectively. Ethylene production from ethanol is based on standard assumptions for Brazil and the USA (Supplementary Table 4). Sensitivity analyses explore assumptions in irrigation (Supplementary Table 5) and nutrient application rates for different yields (Supplementary Table 6), as well as model parameters for biodiversity (Supplementary Table 7).

**LUCI-LCA.** To produce the LUCI-LCA, we first develop predictive land change models to translate the demand scenarios into maps of agricultural expansion and intensification. We then feed the resulting land-use change maps into models for biodiversity and ecosystem services (InVEST) to assess the environmental impacts

of the additional product demand in a spatially explicit way. Finally, we integrate the results to substitute for key elements of the land-use change impacts in standard LCA, as illustrated in Fig. 1 (and described in detail in the Supplementary Methods).

Spatial scenarios of agricultural expansion and intensification are generated from anticipated changes in commodity demand in a region. This approach can be applied with public, globally available data and limited LCM expertise. We derive the potential land area for expansion required to achieve the increase in production based on expansion only (Supplementary Fig. 3 and Supplementary Table 8), then the total land expansion is adjusted to account for intensification by creating a spatially explicit yield map to partition production into amounts met through expansion and intensification (Supplementary Table 9). The total expansion area is then allocated spatially within the region of interest using logistic regressions with maximum likelihood estimators, which identify the combination of parameter values that are most likely to produce the observed data. We use the R package 'lulcc', which provides a workflow to connect raster data to the glm function for generalized linear models. Land cover (MODIS[42]) data from a reference year (in this case, 2007) are reclassified according to binary variables indicating whether each pixel is classified as agriculture. Areas that are assumed to be unable to convert to agriculture (urban, barren and water) are omitted from the regression. The final set of predictor variables selected in the best-fit model include: pH; slope; per cent silt (for Iowa only); and per cent clay and soil organic matter (for Mato Grosso only; Supplementary Tables 10 and 11). We compare actual land-use change that occurred between 2007 and 2012 in both regions (Iowa for maize and sugarcane from Mato Grosso) with the output from the logistic land change model (Supplementary Fig. 4 and Supplementary Table 12), and confirm that findings are plausible based on broader trends for the regions (Supplementary Note 2).

We use ecosystem services modelling (InVEST) to assess impacts from the increased production to meet the different scenario demand targets. The spatially explicit effects of agricultural expansion are modelled in Iowa and Mato Grosso (Supplementary Note 3), for carbon loss (InVEST Carbon Storage and Sequestration and Forest Carbon Edge Effects models; Supplementary Table 13), nitrogen export (InVEST Nutrient Delivery Ratio model; Supplementary Tables 14–16), water consumption from irrigation (InVEST beta model for Blue Water Consumption; Supplementary Tables 17–19), sediment export (InVEST Sediment Delivery Ratio model; Supplementary Tables 20 and 21) and biodiversity (MSA reduction (InVEST GLOBIO model; Supplementary Tables 22 and 23). The spatial resolution of the globally available data used (MODIS land use at 500 m; Digital Elevation Model at 90 m) allows for these spatially explicit modelling approaches to capture finer-scale processes than is possible in standard LCA, such as fragmentation for biodiversity and water routing for sediment and nitrogen export. In addition, the effects of intensification are estimated for nitrogen export and water consumption, via the InVEST Crop Production Model, predicting the relationship between yields, nutrient application and irrigation (Supplementary Notes 4 and 5, and Supplementary Table 15).

We test for model performance and sensitivity of ecosystem service impacts on LCM in two ways (Supplementary Methods). (1) To create a validation layer for the ecosystem service impacts predicted by the LCM, we generate a binary map of where conversion to agriculture occurred between 2007 and 2012, and then overlay these pixels as new agriculture onto the 2007 landscape (Supplementary Fig. 4). We then use the absolute amount of conversion between 2007 and 2012 as a new demand scenario to feed into the LUCI-LCA and calculate the impact per tonne of HDPE. The per-tonne impact can then be compared to the logistic regression model, as shown in Fig. 4. (2) Sensitivity of ecosystem service impacts to LCM are further tested using the InVEST proximity-based scenario generator, which generates an agricultural expansion map for a given area based on distance to or from the frontier of certain land cover classes (in this case, agriculture). When running this scenario generator, we also apply the same restrictions for the types of land that can be converted—specifically, omitting barren, urban and water (Supplementary Fig. 4). The results for the proximity-based LCM are also shown in Fig. 4.

**Integration of LUCI approach into LCA.** In LUCI-LCA, the outputs of the individual InVEST models are used to estimate ecosystem impacts and to directly substitute for key elements (inventory data) in the agricultural stage of standard LCA. For global warming potential, results of spatially explicit modelling substitute for the estimates of carbon dioxide emissions from land-use change. We also consider the spatially explicit impacts of irrigation (Supplementary Tables 24 and 25) and agricultural intensification (Supplementary Tables 26 and 27) on global warming potential. For eutrophication potential, the spatially explicit modelling of nutrient loss substitutes for the values of nitrate leaching from the fields. Intensification and irrigation also affect eutrophication potential. Water consumption considered here is for irrigation only, and thus the InVEST Blue Water Consumption estimate replaces the life cycle inventory for the volume of consumed water during the irrigation process for crop production. The main differences when compared to the LCA data, obtained from the WFN database[36], are the data sources and processing (Supplementary Tables 5 and 28); results are also compared to those obtained using the AWARE methodology (Supplementary Note 6). Likewise, the InVEST method for calculating sediment export and MSA is a direct replacement for the approaches of Saad et al.[6] and De Baan et al.[37] to calculating erosion potential and biodiversity damage potential, respectively, within standard LCA for the agricultural phase of the life cycle.

We test for model sensitivity to parameter error by varying key model parameters (see 'sensitivity' section in each model described in Supplementary Methods). This includes standard errors for biomass estimates for land cover classes from which carbon storage is derived; nitrogen loading for Nutrient Delivery Ratio; quick flow, crop coefficients and total water withdrawals for Blue Water Consumption; C-factors for Sediment Delivery Ratio; and standard errors for MSA estimates for GLOBIO. Sensitivities for LCA (for both the standard approach and on top of model parameter sensitivity already included in the LUCI approach) are further investigated based on uncertainty in the relationships between irrigation, fertilizer application and yields (Supplementary Methods). The effects of pumping water for irrigation are varied using lower and upper irrigation water volumes based on the lower and upper consumed water volumes (Supplementary Table 5), and the N-fertilizer application rates and yields are varied using the lower and upper N-fertilizer application rates (Supplementary Table 6). The sensitivity analysis is summarized in Supplementary Table 29, and presented as error bars on the LUCI-LCA (logistic regression) and standard LCA estimates in Fig. 4.

**Data availability.** All model inputs and outputs and code used to generate the LUCI results are publicly available, archived at http://data.naturalcapital-project.org/LUCI-LCA_chaplin-kramer_et_al_2017. All model inputs and the procedure for deriving the inventories for the standard LCA are publicly available through the references cited and methods described in the Supplementary Methods.

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

## Acknowledgements

We thank Henry King and Jim Leape for valuable comments and perspectives on the approaches developed here. Louis Lindenberg provided guidance on the production scenarios for the case study. The work was funded by grants from Unilever Research and Development, and the Gordon & Betty Moore Foundation.

## Author contributions

R.C.-K. led the work undertaken by the Natural Capital Project team (P.H., B.B., R.N., V.K. and R.S.); S.S. led the work undertaken by the Unilever team (C.M., G.R., M.K. and J.C.); and E.P., S.P., M.R. and G.D. provided conceptual guidance. For LUCI-LCA, P.H. conducted the water consumption analysis and parameterized the sediment and nutrient analyses; R.N. performed the agricultural expansion and intensification analysis and the initial logistic land-change model runs; B.B. performed the uncertainty analysis on the LCM; R.C.-K. parameterized the GLOBIO and carbon analyses, and

developed the crop yield-nutrient application rate relationships; R.S. advised on and adjusted InVEST software; and V.K. generated code and conducted all final InVEST runs. For standard LCA, G.M. and M.K. selected the inventories for ethylene production, C.M. produced MSA and sediment results, G.M. produced nutrient results, M.K. produced carbon results and J.C. produced water consumption results; R.C.-K. and S.S. led the framing and writing of the manuscript; and all authors contributed to manuscript and/or supplement.

## Additional information

**Competing interests:** The authors declare no competing financial interests.

**Publisher's note**: 

