## [Peer Review File · Nature Communications]

Reviewers' Comments:

Reviewer #1 (Remarks to the Author):

This is a thoughtful analysis that I think introduces a much needed advance to LCA of the environmental impacts of product production. My main criticism is that the paper oversells itself regarding the problems that the advances presented actually overcome. Framing the contribution of the paper more humbly and discussing concrete limitations and ways in which the analyses could be advanced would greatly strengthen the paper.

1. Even with LUCI-LCA the estimates of the impact of a particular commodity – and hence the contribution of that impact to an end product – are still taken from averages across huge regions where demand is expected to result in greater expansion and/or intensification (e.g. expansion of soy into the Brazilian cerrado). In reality companies' source from particular sub-regions that, for the reasons outlined in the paper, may vary substantially in environmental characteristics and sustainability criteria.

2. You claim that the approach can be applied using globally available data. However, the dependence on globally available data undermines the importance of the very spatial heterogeneity and landscape effects that you lay out in the introduction. MSA, for example, is a very crude way of assessing the effects of landscape fragmentation and impacts on wider ecosystems processes.

3. The analyses combines - and therefore largely confounds - two very different sources of uncertainty – spatial heterogeneity in production and predictions of demand for a given commodity into the future. It has been well demonstrated that predictions of demand are very hard to make with any certainty, and the fact that the spatial explicitness of your modelling approach depends critically upon the projections of future demand, and not information on spatially explicit sourcing patterns to date, makes this doubly problematic. You argue that the dependence of current methods for estimating LUC relying on linear extrapolations of past trends is a weakness, but this weakness is perhaps more than offset by the uncertainty in how to project demand, including the courageous assumptions that demand will be met by US maize or Brazilian sugarcane.

4. To address some of these limitations in the paper it would be helpful to have a section in the discussion that identifies future improvements and challenges for research. Here (and elsewhere in the paper) more could be done to review other work that is trying other (complementary) ways to link consumption impacts to more fine-grained spatial data on commodity production, e.g. recent work by Godar and colleagues:

Godar, J., Persson, U.M., Tizado, E.J., Meyfroidt, P., 2015. Towards more accurate and policy

relevant footprint analyses: Tracing fine-scale socio-environmental impacts of production to consumption. *Ecol. Econ.* 112, 25–35. doi:10.1016/j.ecolecon.2015.02.003

Godar, J., Suavet, C., Gardner, T.A., Dawkins, E., Meyfroidt, P., 2016. Balancing detail and scale in assessing transparency to improve the governance of agricultural commodity supply chains. *Environ. Res. Lett.* 11, 35015. doi:10.1088/1748-9326/11/3/035015

Flach, R., Ran, Y., Godar, J., Karlberg, L., Suavet, S., 2016. Towards more spatially explicit assessments of virtual water flows: linking local water use and scarcity to global demand of Brazilian farming commodities. <http://iopscience.iop.org/article/10.1088/1748-9326/11/7/075003/pdf>

Minor comments

1. Consequential and attributional LCA need to be defined for the reader
2. Subtitles could be made more meaningful / stand-alone

Reviewer #2 (Remarks to the Author):

The paper claims to provide an improved assessment of the effects of land use change on the LCA for biofuels and bioproducts by using spatially-detailed modeling.

While I strongly support the idea of spatially-explicit modeling to inform LCA of biobased products and fuels, I think this paper does not justify the fundamental assumption that it makes. That assumption is that there will necessarily be land use change associated with increased production of biofuels/bioproducts. Land use change will occur only if the productive capacity of the existing land is inadequate to meet the increased demand due to biofuels or bioproducts. As long as per acre yields are growing faster than demand, then the market can be satisfied using existing crop acres---no land use change will occur.

Yields of maize and sugar cane continue to increase in the U. S. and Brazil, respectively.

<http://usda.mannlib.cornell.edu/MannUsda/viewDocumentInfo.do?documentID=1593>

Thus the increased demand for these feedstocks can, in principle, be met from increased yields on a constant (or even declining) land base. Thus there will not be any land use change if the increased demand is met by increasing yields.

For example, the U. S. maize yields are increasing at about 1% per year. Based on a maize crop of 14 billion bushels per year, assuming 15% moisture content and 80% starch in the dry matter, then this is equivalent to an additional 2.66 million tons per year of starch--without any increase in acres and therefore no land use change. This is approximately 100 times greater than the amount of HDPE that is proposed to be produced.

How can the HDPE demand at this level ever hope to induce land use change, let alone significantly reduce the amount of additional grain produced each year? Unless land use change is demonstrated conclusively, then a model assuming land use change as its underlying assumption is fundamentally flawed.

If the authors would like to rewrite this article showing that land use change is likely or demonstrated to occur, then the article has value. Or if they prefer to heavily caveat their findings by saying repeatedly that this model requires that land use change be proven if the results are to be useful. If not, this is simply an "academic" exercise of little real value.

Some additional concerns/comments.

The data quality is an issue.

- Since national scale data are used in corn and sugarcane (e.g., US corn & Brazil sugarcane in the ecoinvent, FAO data, etc.), I am skeptical that results reflect Iowa or Mato Grosso conditions.
- In the supplemental material, corn in Iowa needs 3.01 cubic meter per ton of blue water. However, generally corn in Iowa does not need irrigation. The irrigated corn acreages are less than 1% of the total.
- What data sources are used in the actual land use change for corn in Iowa? The LUC values in Iowa are quite different from the statistics in the cropland data layer from USDA. According to the cropland data layer from 2010 to 2014, land use change for corn in Iowa is 33% corn acreage from continues corn, 61% from soybean, 1.8% from other annual crops, 2.1% from perennial, and 0.98% from developed lands. Forest conversion to cornfield is about 0.19% of the total corn acreage.
- The assumption that the cropland requirement for expansion in Iowa includes only perennial, grassland, savanna, shrubland and forest needs more references.

<http://usda.mannlib.cornell.edu/MannUsda/viewDocumentInfo.do?documentID=1593>

Reviewer #3 (Remarks to the Author):

The manuscript provides a useful and necessary attempt to enhance the relevance and accuracy of Life Cycle Assessment (LCA) results of bio-based products and systems. Its adequacy is tested with realistic scenarios of agricultural expansion linked to private sector demands of bio-based materials, and results of spatial-modelling-improved LCA results show significant

differences with the results of spatially-unresolved (traditional) LCA. The supplementary material provides concise and clear descriptions of the methods proposed. In my opinion, the manuscript deserves publication with a few changes, mainly related to the consideration of the latest developments in the life cycle impact assessment field: this is because the latest publications in LCA do include further degree of spatial differentiation. The overall results and conclusions reached by the authors are unlikely to be changed, because a significant drive of their results is in the inventory modelling phase of the LCA study, rather than the impact factors considered, but it is important for the manuscript to be up to date at the time of publishing. Another general comment relates to the policy relevance of the discussion. The authors mainly highlight the importance of the findings for the decisions being made by multinational companies; these decisions certainly benefit from the capabilities of the new modelling approaches demonstrated by the authors. However, it would be very useful to also highlight policy decisions that may actually have bigger implications at the landscape level, such as the instalment of incentives to promote specific products (e.g. bio-fuels; bio-materials) or practices (e.g. organic agriculture; good agricultural practices). I would appreciate some brief discussion on how the modelling in InVEST would cope with informing such policy decisions. In summary, this is a bold and useful paper; it really pushes the boundary of the typical LCA applications, and goes a long way in overcoming limitations of standard LCA and providing a solid basis for decision-making on large-scale changes in material sourcing. This could also be very informative in informing environmental implications of policy particularly in the context of the 2030 Agenda for Sustainable Development. The paper has the potential to significantly influence the way Life Cycle Assessment is performed when informing large-scale decisions that have the potential to drive significant changes at the landscape level. The supplementary material is also very useful, and, together with extensive documentations of the models used (particularly InVEST) would allow other researchers to reproduce the work. I would however advise to update 2-3 of the references and models used for the LCA part, as described in the more detailed comments.

More specific remarks include:

- Abstract: recent LCA methodologies actually advance significantly in terms of spatial resolution, thus rather than existing the authors could mention “commonly used” methodologies. In fact the most recent guidance document from the UNEP/SETAC Life Cycle Initiative provides recommendations in the land use impacts on biodiversity and water use impacts on ecosystems allowing spatial resolution to the scale of ecoregions. It can be downloaded from <http://bit.ly/2d3rkNm>. This reference, together with key publications stemming from it such as the description of the AWARE indicator for water, should be considered by the authors as the recommended practice and indicators for LCA.
- Page 2, L.1: please consider adding a reference explaining the difference between attributional and consequential LCA, for the reader’s understanding.
- Page 2, Figure 1: please explain MSA (Mean Species Abundance) in the Figure’s legend.

- Page 3, last paragraph: in fact, the UNEP/SETAC reference mentioned does not prescribe the use of “pristine baseline”, but mentions Potential Natural Vegetation as the most useful reference. Chaudhary et al. indeed use a near to natural local habitat as a reference, and this is also recommended in the latest guidance recently published (referenced above).
- Page 4, last paragraph above figure: “manner” instead of “manor”?
- Page 4, Figure 3: “ton” may cause confusion with the American ton (as opposed to metric ton, or tonne, which I presumed is referred to here)?
- Page 7, description of Standard LCA: I find it odd that the Water Footprint Network method is suggested in this approach, given that the WFN does not really follow LCA methodology. Other methods / guidance could be more relevant here, especially those recommended by wulca.org, and/or the recent guidance of the UNEP/SETAC Life Cycle Initiative already mentioned above (<http://bit.ly/2d3rkNm>). In the same section, the method suggested for biodiversity impacts has also been recently improved from the use of MSA to a corrected version of the Species Area Relationship, namely the Countryside-SAR; this latter approach is used by Chaudhary et al. (2015) in an improvement of the method used by the authors (de Baan et al. (2013). Chaudhary et al. is actually the method recommended by the Life Cycle Initiative and it would thus make more sense for the authors to use it.

Response to Reviewer Comments for:

Predictive, Spatial Modeling for Biodiversity and Ecosystem Services in Life Cycle Assessment

NCOMMS-16-22169

Rebecca Chaplin-Kramer, Sarah Sim, Perrine Hamel, Benjamin Bryant, Ryan Noe, Carina Mueller, Giles Rigarlsford, Michal Kulak, Virginia Kowal, Richard Sharp, Julie Clavreul, Edward Price, Stephen Polasky, Mary Ruckelshaus, and Gretchen Daily

We thank the Editor and the reviewers for the opportunity to improve this manuscript. We found the comments very helpful and believe the resulting product is a more nuanced and clearer description of our findings and their implications. Thank you very much for your time and attention.

Reviewer #1 (Remarks to the Author):

This is a thoughtful analysis that I think introduces a much needed advance to LCA of the environmental impacts of product production. My main criticism is that the paper oversells itself regarding the problems that the advances presented actually overcome. Framing the contribution of the paper more humbly and discussing concrete limitations and ways in which the analyses could be advanced would greatly strengthen the paper.

We thank the reviewer for these helpful suggestions in setting the appropriate tone and scope for the paper. We have scaled back some of the assertions to better reflect the contribution of this approach, and the work still needed to be done.

1. Even with LUCI-LCA the estimates of the impact of a particular commodity – and hence the contribution of that impact to an end product – are still taken from averages across huge regions where demand is expected to result in greater expansion and/or intensification (e.g. expansion of soy into the Brazilian cerrado). In reality companies' source from particular sub-regions that, for the reasons outlined in the paper, may vary substantially in environmental characteristics and sustainability criteria.

It is certainly true that the lack of traceability for many currently sourced commodities poses a major problem for companies in understanding their environmental impacts, because the raw material production stage often incurs the largest share of a product's impact (Nemecek et al. 2016 Int. J. Life Cycle Assess. 21, 607–620). Knowing the location of raw material production is thus very important to understanding the full impact. Further compounding this uncertainty, our work is positioned as predictive – focused on helping companies understand what impacts could or will likely occur as a result of a product innovation or of future sourcing decisions. By definition, in these cases, one would not know the likely source of materials without additional modeling. Our aim here is to show – for some illustrative scenarios of possible materials sourcing – how to: (i) frame and quantify their potential impacts; (ii) differentiate between sourcing regions in terms of the impacts assessed; and (iii) understand underlying drivers of the impacts more fully. Our results are the aggregated effects of spatially-explicit impacts – based on the pixels of likely conversion, rather than the entire sourcing region. We have revised the introduction and discussion to make clearer the intended contribution of this effort, specifically and in the greater context.

2. You claim that the approach can be applied using globally available data. However, the dependence on globally available data undermines the importance of the very spatial heterogeneity and landscape effects

that you lay out in the introduction. MSA, for example, is a very crude way of assessing the effects of landscape fragmentation and impacts on wider ecosystems processes.

Key point. Yet “globally available” does not always mean too coarse to be informative! MODIS provides 500 m land-use data, and Landsat has spurred the creation of several 30 m forest cover data sets, which will likely be improved to include additional land covers/uses (such as in agriculture) in the near future. These globally available but fine-scale data are adequate for detecting broad trends in fragmentation and landscape configuration. Indeed, quantification of biodiversity impacts in standard LCA practice does ignore fragmentation, considering only land uses and areal extents, whereas our approach as computed in InVEST includes fragmentation at a scale relevant to many species (using the 500 m MODIS data).

3. The analyses combines - and therefore largely confounds - two very different sources of uncertainty – spatial heterogeneity in production and predictions of demand for a given commodity into the future. It has been well demonstrated that predictions of demand are very hard to make with any certainty, and the fact that the spatial explicitness of your modelling approach depends critically upon the projections of future demand, and not information on spatially explicit sourcing patterns to date, makes this doubly problematic. You argue that the dependence of current methods for estimating LUC relying on linear extrapolations of past trends is a weakness, but this weakness is perhaps more than offset by the uncertainty in how to project demand, including the courageous assumptions that demand will be met by US maize or Brazilian sugarcane.

We certainly agree with this assessment of uncertainty in predictions of demand – but the demand scenarios in this exercise are not intended as predictions. Rather, they are scenarios meant to illustrate decisions taken by a company like Unilever. The predictive element of our analysis is in exploring how possible (scenario) increases in future demand will be met through changes in land-use, and these predictions are based on land suitability determined by existing spatially explicit production patterns. While we do have an assessment of uncertainty in the spatially-explicit land-use change within a region (by comparing to historic agricultural expansion), we do not account for the uncertainty that the chosen focal regions represent in terms of where future demand will be met because this is beyond the scope of the current exercise. Ultimately this approach should be coupled with econometric approaches, to narrow down the regions from which increased demand for specific commodities would likely be met, and then to use LUCI-LCA in each of these regions.

We have added a paragraph to the introduction (L 46-55) to clarify that modeling impacts of a decision-relevant shift in commodity demand is a three-step process: 1) determine the location (country / state) and amount (ha) of LUC (intensification and expansion) induced by the demand shift 2) distribute the demand across the landscape using evidence-based criteria (in this case suitability and proximity) 3) estimate the potential impacts. We have done 2) and 3) – and we now more clearly highlight the gap in step 1 in the discussion (L 199-216).

As a placeholder for step 1) in this approach, we take two sourcing regions, by way of illustration, and use FAO statistics to partition the demand volume into the amount met by intensification vs. expansion. In our specific case study, the assumption of US and Brazil and the particular states chosen within them is a means of demonstrating the capabilities we have developed in 2) and 3). The choice of regions is not completely unfounded, however – the focal countries and states are where the main commercial investments for these feedstocks and associated plastics production are currently being made (Bioplastics Council 2012).

Bioplastics Industry Overview Guide, Executive Summary Report, The Society of the Plastics Industry, Inc. <http://www.plasticsindustry.org/files/about/BPC/Industry%20Overview%20Guide%20Executive%20Summary%20-%200912%20-%20Final.pdf>

4. To address some of these limitations in the paper it would be helpful to have a section in the discussion that identifies future improvements and challenges for research. Here (and elsewhere in the paper) more could be done to review other work that is trying other (complementary) ways to link consumption impacts to more fine-grained spatial data on commodity production, e.g. recent work by Godar and colleagues:

Godar, J., Persson, U.M., Tizado, E.J., Meyfroidt, P., 2015. Towards more accurate and policy relevant footprint analyses : Tracing fine-scale socio-environmental impacts of production to consumption. *Ecol. Econ.* 112, 25–35. doi:10.1016/j.ecolecon.2015.02.003

Godar, J., Suavet, C., Gardner, T.A., Dawkins, E., Meyfroidt, P., 2016. Balancing detail and scale in assessing transparency to improve the governance of agricultural commodity supply chains. *Environ. Res. Lett.* 11, 35015. doi:10.1088/1748-9326/11/3/035015

Flach, R., Ran, Y., Godar, J., Karlberg, L., Suavet, S., 2016. Towards more spatially explicit assessments of virtual water flows: linking local water use and scarcity to global demand of Brazilian farming commodities. <http://iopscience.iop.org/article/10.1088/1748-9326/11/7/075003/pdf>

We thank the reviewer for sharing these citations; this is indeed exciting work and we have added a discussion of how these advances could improve the approach in L 203-208.

Minor comments

1. Consequential and attributional LCA need to be defined for the reader

Thank you. A brief parenthetical added in L 42-43 articulates the difference between the two. This is kept brief in the main text so as not to break up the flow, but references a section of the supplement (L 44-45) where more detail that has now been added.

2. Subtitles could be made more meaningful / stand-alone

Great suggestion. We have lengthened or revised each of the subtitles to clarify their message; this does put us over the 60 character limit specified in the document formatting requirements, so will let the Editor determine whether we are able to include these more meaningful subtitles.

Reviewer #2 (Remarks to the Author):

The paper claims to provide an improved assessment of the effects of land use change on the LCA for biofuels and bioproducts by using spatially-detailed modeling.

While I strongly support the idea of spatially-explicit modeling to inform LCA of biobased products and fuels, I think this paper does not justify the fundamental assumption that it makes. That assumption is that there will necessarily be land use change associated with increased production of biofuels/bioproducts. Land use change will occur only if the productive capacity of the existing land is inadequate to meet the increased demand due to biofuels or bioproducts. As long as per acre yields are growing faster than demand, then the market can be satisfied using existing crop acres---no land use change will occur.

We understand the concern here and have revised the manuscript substantially in response. We believe, however, that our fundamental premise is acceptable, for three reasons:

- 1) Our analysis is meant to demonstrate a general approach, and the scenarios we used are meant to be illustrative rather than decision-ready. Our study aim is to extend the impacts assessment of LUC in LCA with improved, more spatially-explicit and ecologically-driven methods. We now clearly acknowledge that we are using the scenarios of the amount of LUC required by an increase in commodity demand as a place-holder, because our focus is on improving the impacts*

of that LUC, and have added paragraphs both in the introduction and discussion to highlight the recent advances and the need for continued advances in supply chain transparency and econometric modeling that can better predict how and where an increase in demand would translate to LUC.

- 2) We do define LUC as both expansion AND intensification, and both of these are associated with environmental trade-offs that we account for in our approach. In fact, our estimate of how much of future demand could be met by intensification is if anything overestimated because the calculation was based on linear extrapolation of past yield increases, and there is evidence that yields in many regions of the world are plateauing (Ray et al. 2013 PloS one 8.6: e66428). Furthermore, even if yield increases are technically possible, they are often hindered by a range of economic, social and governance factors, so agricultural expansion continues to be likely in many countries (Mandemaker et al. 2011 Ecology & Society), and can even be further spurred by intensification if it is market-driven rather than technology-driven and well-regulated (Byerlee et al. 2014 Global Food Security 3: 92-98).*
- 3) The argument involving yields growing faster than any given demand shock is missing the point, because decisions (such as modeled here) are not happening in a vacuum where everything else stays the same. Globally, future population and consumption increases are expected to drive a doubling of agricultural demand by 2050 – which far exceeds production that yield increases alone can meet (Ray et al. 2013 PloS one 8.6: e66428). These megatrends are the backdrop against which individual decisions need to be considered; demand is outpacing supply in general, even without the additional increased demand that would occur if companies made the decision to switch from petro-based to bio-based plastics..*

Yields of maize and sugar cane continue to increase in the U. S. and Brazil, respectively.

<http://usda.mannlib.cornell.edu/MannUsda/viewDocumentInfo.do?documentID=1593>

See point 2 above, and Fig S3 in Supplemental Materials. We do include this trend of yield increases in our approach.

Thus the increased demand for these feedstocks can, in principle, be met from increased yields on a constant (or even declining) land base. Thus there will not be any land use change if the increased demand is met by increasing yields. For example, the U. S. maize yields are increasing at about 1% per year. Based on a maize crop of 14 billion bushels per year, assuming 15% moisture content and 80% starch in the dry matter, then this is equivalent to an additional 2.66 million tons per year of starch--without any increase in acres and therefore no land use change. This is approximately 100 times greater than the amount of HDPE that is proposed to be produced.

Despite these yield increases, there has been significant land-use change in both countries. We have added a new section to the supplement (SM 3.5.2) documenting the trend in recent decades, for Mato Grosso and for Iowa. Recent studies have found that southern Iowa counties, in particular, have transitioned from non-ag to corn (Stern et al., 2012 J. Appl. Remote Sens. 6, 63590), and that croplands have “substantially infilled the lesser-cultivated areas of Southern Iowa, a region characterized by steeply sloped hills normally reserved for livestock grazing” (Lark et al., 2015 Environ. Res. Lett 10, 44003)

How can the HDPE demand at this level ever hope to induce land use change, let alone significantly reduce the amount of additional grain produced each year? Unless land use change is demonstrated conclusively, then a model assuming land use change as its underlying assumption is fundamentally flawed.

See above; increased demand cannot be expected to be met by intensification alone, and has not been, according to past study.

If the authors would like to rewrite this article showing that land use change is likely or demonstrated to occur, then the article has value. Or if they prefer to heavily caveat their findings by saying repeatedly that this model requires that land use change be proven if the results are to be useful. If not, this is simply an "academic" exercise of little real value.

As noted above, the evidence is very strong that the megatrends of growing population and consumption are going to induce LUC of some kind— a mix of intensification and expansion. Our goal is to advance systematic ways of evaluating the impacts of future change. Our analysis examines the differences in outcomes between standard LCA and our new approach in response to a hypothetical but realistic demand-shock scenario of a large company.

Some additional concerns/comments.

The data quality is an issue.

- Since national scale data are used in corn and sugarcane (e.g., US corn & Brazil sugarcane in the ecoinvent, FAO data, etc.), I am skeptical that results reflect Iowa or Mato Grosso conditions.

To keep our approach consistent with standard LCA, we prioritized globally consistent data to maintain consistency between study regions, at the expense of local precision. The national data were only used to set up the scenario; the modeling itself was done with data at resolutions ranging from 90 m to 1 km (with 500 m land use data; MODIS).

- In the supplemental material, corn in Iowa needs 3.01 cubic meter per ton of blue water. However, generally corn in Iowa does not need irrigation. The irrigated corn acreages are less than 1% of the total.

Based on our sources, corn production in Iowa is associated with some water irrigation. We have compiled evidence for this in sections 3.2.3.3.1 and 3.2.3.3.2 of the Supplemental Material, where we compare our irrigation estimates (obtained from the water use model) to other data sources. In Iowa, these include USGS data (Ref 67) and the Water Footprint database (Ref. 10). Additional estimates can be found in a study by Chiu et al. (2009) (ES&T 43:2688-2692): they show for Iowa a total volume of irrigation water of 17,288 million liters for 2007 for an ethanol production from corn of 6.857 million liters. This equates to 2.57 L irrigation water per L ethanol. Using a rough yield of 3 gallons per bushel corn (<http://www.eia.gov/todayinenergy/detail.php?id=21212>) we get a water consumption of 1.4 m³ per ton corn. This is close to the data we use in this study (from the WFN database) which is 1.65 m³ water per ton corn (consumed volume of water – rather than the irrigation blue water volume, which incorporates the irrigation efficiency losses and is equal to 3.01 m³ water per ton corn).

- What data sources are used in the actual land use change for corn in Iowa? The LUC values in Iowa are quite different from the statistics in the cropland data layer from USDA. According to the cropland data layer from 2010 to 2014, land use change for corn in Iowa is 33% corn acreage from continues corn, 61% from soybean, 1.8% from other annual crops, 2.1% from perennial, and 0.98% from developed lands. Forest conversion to cornfield is about 0.19% of the total corn acreage.

MODIS was used for calculating actual land use change from 2007-2012, and this is now clarified in the supplemental methods section 3.1.3.3. The best data for Iowa (from the ag census) show increases in cropland and decreases in woodland and pastureland; see tables added to SM 3.5.2 (appendix II) and pasted in comment below. These census data confirm the trend we saw in MODIS-derived data. We used MODIS, rather than the Cropland Data Layer (CDL), because we are constructing an approach for global replicability. Furthermore, the CDL isn't reliable at identifying natural vegetation like forest or grassland, and its non-ag training data is from the 2011 National Land Cover Dataset (which is actually 2009-2011 imagery). Because the CDL training data are not ground-truthed, the CDL does not incorporate an

accuracy assessment of the classification of non-cropland, which we consider more important to our assessment than the classification of corn vs. other crops. While MODIS-derived data are not crop specific, meaning we have no estimates on what transitioned to or from corn specifically, the CDL suggests that the scenarios we use are plausible. The figure the reviewer provided of 0.19% of total corn land in Iowa is ~10,000 hectares- which is in line with our scenarios.

- The assumption that the cropland requirement for expansion in Iowa includes only perennial, grassland, savanna, shrubland and forest needs more references.

<http://usda.mannlib.cornell.edu/MannUsda/viewDocumentInfo.do?documentID=1593>

Iowa has lost pasture/grassland and woodland while it has gained cropland over the past two decades (see table below; this has been reformatted as percent of total land area in Iowa and added, along with similar figures for Mato Grosso, to SM 3.5.2 - Appendix II in supplement). This demonstrates conclusively that the trends shown in the scenarios (as well as the actual land-use change from agricultural expansion given by MODIS) falls in line with the USDA records for IA between 2007 and 2012.

USDA Agricultural Census data for Iowa (acres)

Program	Year	Pastureland and pastured cropland	Woodland (including pastured woodlands)	Cropland (excluding pastured cropland)
CENSUS	1997	3,592,036	1,406,516	25,645,673
CENSUS	2002	3,090,582	1,336,752	25,798,130
CENSUS	2007	2,744,708	1,193,303	25,486,548
CENSUS	2012	2,130,373	1,165,549	26,032,384

Source: USDA-NASS, 2015. National Agricultural Statistics Services QuickStats Ad-hoc Query Tool.

URL <http://quickstats.nass.usda.gov/>

Reviewer #3 (Remarks to the Author):

The manuscript provides a useful and necessary attempt to enhance the relevance and accuracy of Life Cycle Assessment (LCA) results of bio-based products and systems. Its adequacy is tested with realistic scenarios of agricultural expansion linked to private sector demands of bio-based materials, and results of spatial-modelling-improved LCA results show significant differences with the results of spatially-unresolved (traditional) LCA. The supplementary material provides concise and clear descriptions of the methods proposed. In my opinion, the manuscript deserves publication with a few changes, mainly related to the consideration of the latest developments in the life cycle impact assessment field: this is because the latest publications in LCA do include further degree of spatial differentiation. The overall results and conclusions reached by the authors are unlikely to be changed, because a significant drive of their results is in the inventory modelling phase of the LCA study, rather than the impact factors considered, but it is important for the manuscript to be up to date at the time of publishing.

We appreciate these valuable points. We now include a paragraph on these recognized advances, the AWARE characterization factors for water and the Chaudhary approach to biodiversity, in L 175-188; see comment below for more detail.

Another general comment relates to the policy relevance of the discussion. The authors mainly highlight the importance of the findings for the decisions being made by multinational companies; these decisions certainly benefit from the capabilities of the new modelling approaches demonstrated by the authors. However, it would be very useful to also highlight policy decisions that may actually have bigger implications at the landscape level, such as the instalment of incentives to promote specific products (e.g. bio-fuels; bio-materials) or practices (e.g. organic agriculture; good agricultural practices). I would appreciate some brief discussion on how the modelling in InVEST would cope with informing such policy decisions.

We thank the reviewer for pointing this out; it is very true that other landscape decisions stand to impact biodiversity and ecosystem services greatly, and indeed most InVEST applications have been in the local and regional policy realm (Ruckelshaus et al. 2015 Ecological Economics 115:11–21). The point of this paper is to show that, although such spatially-explicit modeling is not typically being used in global decision contexts such as corporate supply chains, the processes represented by such modeling are critical to include in such assessments. However, we appreciate that some of the audiences we are trying to reach (especially in the corporate and LCA communities) may not be aware of the approaches taken in other decision contexts, so we have added a brief description of these to L 222-225.

In summary, this is a bold and useful paper; it really pushes the boundary of the typical LCA applications, and goes a long way in overcoming limitations of standard LCA and providing a solid basis for decision-making on large-scale changes in material sourcing. This could also be very informative in informing environmental implications of policy particularly in the context of the 2030 Agenda for Sustainable Development. The paper has the potential to significantly influence the way Life Cycle Assessment is performed when informing large-scale decisions that have the potential to drive significant changes at the landscape level.

We are grateful for this encouraging comment and do hope to have exactly such impact.

The supplementary material is also very useful, and, together with extensive documentations of the models used (particularly InVEST) would allow other researchers to reproduce the work. I would however advise to update 2-3 of the references and models used for the LCA part, as described in the more detailed comments.

We thank the reviewer for these additional references that shed light on more recent developments, and have harmonized our discussion with these recent advances.

More specific remarks include:

- Abstract: recent LCA methodologies actually advance significantly in terms of spatial resolution, thus rather than existing the authors could mention “commonly used” methodologies. In fact the most recent guidance document from the UNEP/SETAC Life Cycle Initiative provides recommendations in the land use impacts on biodiversity and water use impacts on ecosystems allowing spatial resolution to the scale of ecoregions. It can be downloaded from <http://bit.ly/2d3rkNm>. This reference, together with key publications stemming from it such as the description of the AWARE indicator for water, should be considered by the authors as the recommended practice and indicators for LCA.

We have changed the language in the abstract to “commonly used” and include a paragraph on some of these recent advances, as noted in the comment above.

We appreciate the point about including these additional advances. The reference link provided here unfortunately links to a file that says “pre-publication preview” and does not include the section on methods. However, we were able to obtain details on the AWARE method (see below), and have added

a secondary analysis to the supplement that weights the water consumption results by the AWARE characterization for remaining water availability. It does not change our overall results, and we note this in L 181-187, and have added more details on the analysis from which we draw this conclusion to the supplement (SM section 3.5.6).

The publication detailing the final AWARE methodology has not yet been accepted. It is in review at ES&T:

Anne-Marie Boulay, Jane Bare, Lorenzo Benini, Markus Berger, Michael J. Lathuilière, Alessandro Manzardo, Manuele Margni, Masaharu Motoshita, Montserrat Núñez, Amandine Valerie Pastor, Bradley Ridoutt, Taikan Oki, Sebastian Worbe, Stephan Pfister: The WULCA consensus characterization model for water scarcity footprints: Assessing impacts of water consumption based on available water remaining (AWARE). Environmental Science & Technology.

We will be citing the above unless the editor prefers not to cite work in review. In that case, a previous publication by the same group is below, which outlines the process of developing AWARE (but not the detail for the final AWARE methodology):

Boulay A-M, Bare J, De Camillis C, Döll P, Gassert F, Gerten D, Margni M, et al. 2015. Consensus building on the development of a stress-based indicator for lca-based impact assessment of water consumption: Outcome of the expert workshops. The International Journal of Life Cycle Assessment:1-7.

The AWARE methodology and data are available at <http://wulca-waterlca.org/project.html>

- Page 2, L.1: please consider adding a reference explaining the difference between attributional and consequential LCA, for the reader's understanding.

We have added a reference in the main text, as well as a more lengthy explanation in the SM 3.5.1

- Page 2, Figure 1: please explain MSA (Mean Species Abundance) in the Figure's legend.

Done – thank you.

- Page 3, last paragraph: in fact, the UNEP/SETAC reference mentioned does not prescribe the use of “pristine baseline”, but mentions Potential Natural Vegetation as the most useful reference. Chaudhary et al. indeed use a near to natural local habitat as a reference, and this is also recommended in the latest guidance recently published (referenced above).

Poor language choice on our part, that we have revised accordingly in L 110. By “pristine baseline” we in fact meant Potential Natural Vegetation. The point we're trying to make is that our approach uses the current land use as the baseline and looks at additional impacts caused by future land-use change. When using Potential Natural Vegetation, places that are already quite impacted will appear to be disproportionately impacted by future decisions.

- Page 4, last paragraph above figure: “manner” instead of “manor”?

Fixed

- Page 4, Figure 3: “ton” may cause confusion with the American ton (as opposed to metric ton, or tonne, which I presumed is referred to here)?

Changed throughout

• Page 7, description of Standard LCA: I find it odd that the Water Footprint Network method is suggested in this approach, given that the WFN does not really follow LCA methodology. Other methods / guidance could be more relevant here, especially those recommended by wulca.org, and/or the recent guidance of the UNEP/SETAC Life Cycle Initiative already mentioned above (<http://bit.ly/2d3rkNm>). In the same section, the method suggested for biodiversity impacts has also been recently improved from the use of MSA to a corrected version of the Species Area Relationship, namely the Countryside-SAR; this latter approach is used by Chaudhary et al. (2015) in an improvement of the method used by the authors (de Baan et al. (2013). Chaudhary et al. is actually the method recommended by the Life Cycle Initiative and it would thus make more sense for the authors to use it.

A direct comparison with the latest biodiversity CF proposed in Chaudhary et al. utilizing the Countryside Species Area Relationship was not possible, as they have used species numbers as an indicator, while our indicator included not only number of species but also their abundance – combined in the Mean Species Abundance. Consensus in the LCA community has recently emerged (Teixeira et al. 2016 Journal of Cleaner Production 112: 4283-4287) that not just the species number but also their abundance should be considered in a more holistic way to consider impacts on biodiversity. However, we recognize the importance of this advance to the LCA community and the relevance of it to our work, and have added more discussion on this point to L 175-181. Thank you.

We did not use the Water Footprint Network (WFN) method per se but we used the water consumption data provided by the WFN as it provides the best data for water consumption from crop production (e.g., see discussion by Pfister et al. (2014 J. Clean. Prod. 73: 52–62) on the calculation of water consumption for LCAs). For this reason, many LCA practitioners also use the WFN data for their inventories (e.g. Motoshita et al. 2014 Int J of Life Cycle Assessment: 1-12., Lutter et al. 2016 Global Environmental Change 38: 171-182). However, we do now include a comparison of the results when weighted by the AWARE factors, as noted above.

Reviewer Comments:

Reviewer #1 (Remarks to the Author):

I am satisfied with the detailed revision undertaken by the authors. I think the paper is an articulate and innovative contribution that can greatly improve LCA work. My main concern with the initial submission was that the paper oversold its contribution, and in doing so undermined its own value and clarity around what needs to be done next. I think the authors have now addressed this nicely and the paper is much better situated in a broader context. Well done!

Reviewer #2 (Remarks to the Author):

I believe my comments and concerns about the manuscript have been adequately addressed in the rewrite. I recommend this manuscript be published substantially as is.

Reviewer #3 (Remarks to the Author):

I confirm that the points made by me in the previous round have been more than satisfactorily addressed in this revised version of the manuscript. While addressing the comments of the other reviewers, the authors have also improved the paper overall. I think this is now acceptable for publication.

Kind regards

Response to reviews:

We thank the reviewers for their final comments and would like to mention again that we are extremely grateful for their very insightful and helpful suggestions that we believe have much improved the manuscript. We are delighted that they agree.

Reviewer #1 (Remarks to the Author):

I am satisfied with the detailed revision undertaken by the authors. I think the paper is an articulate and innovative contribution that can greatly improve LCA work. My main concern with the initial submission was that the paper oversold its contribution, and in doing so undermined its own value and clarity around what needs to be done next. I think the authors have now addressed this nicely and the paper is much better situated in a broader context. Well done!

Reviewer #2 (Remarks to the Author):

I believe my comments and concerns about the manuscript have been adequately addressed in the rewrite. I recommend this manuscript be published substantially as is.

Reviewer #3 (Remarks to the Author):

I confirm that the points made by me in the previous round have been more than satisfactorily addressed in this revised version of the manuscript. While addressing the comments of the other reviewers, the authors have also improved the paper overall. I think this is now acceptable for publication.
Kind regards